# Haploinsufficiency as a Foreground Pathomechanism of Poirer-Bienvenu Syndrome and Novel Insights Underlying the Phenotypic Continuum of *CSNK2B*-Associated Disorders

**DOI:** 10.3390/genes14020250

**Published:** 2023-01-18

**Authors:** Mariateresa Di Stazio, Caterina Zanus, Flavio Faletra, Alessia Pesaresi, Ilaria Ziccardi, Anna Morgan, Giorgia Girotto, Paola Costa, Marco Carrozzi, Adamo P. d’Adamo, Luciana Musante

**Affiliations:** 1Institute for Maternal and Child Health—IRCCS “Burlo Garofolo”—Trieste, 34137 Trieste, Italy; 2Department of Medical, Surgical and Health Sciences, University of Trieste, 34127 Trieste, Italy

**Keywords:** WES, POBINDS, mRNA and protein instability, CK2beta haploinsufficiency, IDCS, KEN like-box, phenotypic continuum

## Abstract

*CSNK2B* encodes for the regulatory subunit of the casein kinase II, a serine/threonine kinase that is highly expressed in the brain and implicated in development, neuritogenesis, synaptic transmission and plasticity. De novo variants in this gene have been identified as the cause of the Poirier-Bienvenu Neurodevelopmental Syndrome (POBINDS) characterized by seizures and variably impaired intellectual development. More than sixty mutations have been described so far. However, data clarifying their functional impact and the possible pathomechanism are still scarce. Recently, a subset of *CSNK2B* missense variants affecting the Asp32 in the KEN box-like domain were proposed as the cause of a new intellectual disability-craniodigital syndrome (IDCS). In this study, we combined predictive functional and structural analysis and in vitro experiments to investigate the effect of two *CSNK2B* mutations, p.Leu39Arg and p.Met132LeufsTer110, identified by WES in two children with POBINDS. Our data prove that loss of the CK2beta protein, due to the instability of mutant *CSNK2B* mRNA and protein, resulting in a reduced amount of CK2 complex and affecting its kinase activity, may underlie the POBINDS phenotype. In addition, the deep reverse phenotyping of the patient carrying p.Leu39Arg, with an analysis of the available literature for individuals with either POBINDS or IDCS and a mutation in the KEN box-like motif, might suggest the existence of a continuous spectrum of *CSNK2B*-associated phenotypes rather than a sharp distinction between them.

## 1. Introduction

Casein kinase (CK2) is a ubiquitous serine/threonine kinase that is highly conserved in evolution. The constitutively active holoenzyme is composed of two catalytic subunits (alpha and/or alpha’), which associate with a dimeric building block of regulatory beta subunits [1]. The subunits are encoded by *CSNK2A1*, *CSNK2A2* and *CSNK2B*, respectively. Accumulated biochemical and genetic evidence indicates that the enzyme phosphorylates serine and threonine residues of a vast array of target proteins involved in several signaling pathways (e.g., PI3K/Akt, IKK/NFkB, WNT/β-catenin, JAK2/STAT3), thereby regulating various crucial cellular processes, including cell survival, proliferation, differentiation, migration and cell cycle progression [2]. CK2 is especially abundant in the brain with several substrates, mostly proteins related to transcription-directed signaling [3], and it is implicated in development, neuritogenesis, synaptic transmission and plasticity [4]. CK2 activity is fine-regulated by the beta subunit, which mediates the formation of the tetrameric complex and plays a role in stabilizing the catalytic subunits and in modulating the substrate specificity of CK2, its architecture being consistent with its role as a docking partner for other interacting proteins [1,5]. In vitro studies demonstrated that increased expression of the CK2beta protein is sufficient to increase CK2 activity [6], which is negatively regulated in the cells through CK2beta degradation via a proteasome-mediated pathway [7]. In vivo evidence indicates a crucial function of CK2beta in the nervous system; the knockout of CK2beta in mice (Csnk2−/−) results in post-implantation lethality, suggesting an essential role in cell viability [8], whereas conditional Csnk2b knockout mice showed impaired proliferation and differentiation of embryonic neural stem cells in the telencephalon [9]. Importantly, knockdown of Csnk2b in mouse embryonic stem cells (day 14.5) resulted in an altered neuronal morphology, with a reduced number and length of dendrites and defective synaptic transmission [10].

In humans, *de novo* mutations in the *CSNK2B* gene have been recently identified as the molecular cause of the Poirer-Bienvenu Neurodevelopmental Syndrome (POBINDS; MIM: #618732) [11]. The condition can manifest with a broad range of symptoms, including developmental delay (DD), variable degree of intellectual disability (ID) (from mild to severe), language deficits, epilepsy (ranging from well controlled to refractory seizures) and growth abnormalities. Published clinical descriptions are mainly focused on the neurodevelopmental phenotype, but merging data has recently evidenced that dysmorphism and abnormalities of cranial size might represent a common finding in POBINDS patients [12]. In general, a great inter-individual variability of heterogeneous mild dysmorphic features is reported [13]. Nevertheless, in 2022, Asif described digital anomalies and facial characteristics, including deep-set eyes, hypoplastic alae nasi, a broad and depressed nasal bridge, thin lips, prognatism and a pointed chin, which, according to the author, compose a “strikingly similar facial gestalt” in a subset of three patients with discrete *CSNK2B* variants. The authors suggested that the pathogenic variants affecting p.Asp32 lead to upregulation of *CSNK2B* expression at the transcript and protein level, along with global dysregulation of canonical Wnt signalling, and through a dominant negative effect, causing a new intellectual disability-craniodigital syndrome (IDCS) distinct from POBINDS [14]. Since the first description, more than 60 patients have been reported with a variety of *CSNK2B* alterations, including missense, frameshift, nonsense, small indels, start loss and splice site, but the variant type seems not to reflect the clinical severity (HGMD professional database). Although a genotype-phenotype correlation has not yet been established, Zhang and colleagues recently proposed that individuals with variants affecting the zinc-finger domain present a milder phenotype [13].

However, to date, the functional consequences of the *CSNK2B* variants associated with POBINDS, as well as the extent to which they contribute to the physiopathology of the disease, remain unclear.

The current study aims (1) to detail the clinical manifestations and the molecular findings in two patients with *de novo CSNK2B* variants, the first carrying the c.T116G (p.Leu39Arg) variant, affecting the amino acid in position 39 in the KEN box-like domain closed to the arginine 32 mutated in the IDCS patients, and the second with a frameshift deletion (c.384_394delAGGTGAAGCCA) in exon 6, which encodes for the zinc-finger domain; (2) to assess the functional consequences of the POBINDS-associated variants identified in the two patients on *CSNK2B* mRNA, protein structure, stability, cellular localization and kinase activity.

We provide the deep phenotyping of the two patients, the detailed molecular characterization and the functional analysis of POBINDS-associated variants. Our data prove that loss of the CK2beta protein, due to increased instability of mRNA or protein, resulting in a decreased amount of CK2 complex impairing its kinase activity, may underlie the POBINDS phenotype. Our report, which includes an analysis of the available literature concerning patients with a mutation in the KEN box-like (either IDCS or POBINDS), might suggest the existence of a continuous spectrum of *CSNK2B*-associated phenotypes rather than a sharp distinction between them.

## 2. Materials and Methods

### 2.1. Patients

Patients were recruited in the Child Neuropsychiatry Unit of the Pediatric Department of the “IRCCS” Burlo Garofolo in Trieste, Italy. A multidisciplinary team assessed the clinical presentation at disease onset and follow-up. Electroclinical data, radiological data, neuroimaging, cognitive and behavioral tests, dysmorphological features and anthropometric data were collected and evaluated. We compiled the family history, assessed the relatives’ phenotyping and, after genetic counselling, the patients were enrolled for molecular diagnostic purposes. The clinical information was standardized using the Human Phenotype Ontology (HPO) [15] to allow for a reliable comparison of clinical features among individuals. The diagnostic procedure included discussion of WES data in the context of phenotypic data at interdisciplinary meetings [16]. Investigations were conducted according to the Declaration of Helsinki principles. Informed consent was collected from the enrolled subjects.

### 2.2. WES, Interpretation, Validation and In Silico Functional Analysis

The individuals who were negative for pathogenic micro-rearrangements (SNP array analysis) were enrolled for WES analysis. Genomic DNA was extracted from venous peripheral blood lymphocytes using the QIAsymphony^®^ SP instrument with the QIAsymphony^®^ DNA Midi (Qiagen, Hilden, Germany) according to manufacturer’s instructions. WES sequencing was performed on the trio (patient and both parents) at the “IRCCS” Burlo Garofolo (Trieste, Italy). For patient 1, WES methods and data processing, including sequence alignment to the human reference genome, variant filtering and prioritization by allele frequency, predicted functional impact, inheritance and validation, have been previously reported [16]. A summary of the WES method and statistics is presented in Appendix A. For patient 2, DNA was processed for library enrichment with the Twist Human Core for Enrichment-Exome panel (Twist) according to the manufacturer. Sequencing was performed on a NextSeq 500 (Illumina, San Diego, CA, USA) instrument. Sequencing reads were aligned to the human reference genome (GRCh37/hg19) employing the BWA-mem tool, and variant calling was performed with a GATK v4.1.2 HaplotypeCaller (EnGenome, Pavia, Italy). Annotation and prioritization of the variants were conducted with eVAI-enGenome software based on the American College of Medical Genetics and Genomics (ACMG) guidelines [17]. SNVs and INDELs were filtered referring to public databases (dbSNP build150, NHLBI Exome Sequencing Project (ESP), Exome Aggregation Consortium (ExAC) database, Genome Aggregation Database (gnomAD)) and led to ruling out those variants previously reported as polymorphism. In particular, a minor allele frequency (MAF) cut-off of ≤0.01% was utilized. The trio was examined for the following inheritance patterns: *de novo*, homozygous recessive, compound heterozygous and hemizygous.

The variants’ pathogenicity was also assessed with the ClinVar (https://www.ncbi.nlm.nih.gov/clinvar/) accessed on 1 July 2022, the Human Gene Mutation Database professional (HGMD) (Qiagen), Online Mendelian Inheritance in Man (OMIM) (https://www.omim.org/) and DECIPHER (https://www.deciphergenomics.org) accessed on 1 July 2022. Several in silico tools, such as PaPI [18], PolyPhen-2 [19], Sorting Intolerant from Tolerant (SIFT) [20], MutationTaster [21], DANN [22], dbscSNV score [23] and Combined Annotation Dependent Depletion (CADD) score [24], were applied to establish the pathogenicity of novel variants. Cross-species alignment was obtained using ClustalW (https://www.genome.jp/tools-bin/clustaw) accessed on 29 July 2022. The 3D structure of the mutant protein was built employing the SWISS-Model, an automated protein structure homology-modeling server [25], based on the wild-type crystal structure (PDB:1JWH) (https://swissmodel.expasy.org/) accessed on 1 July 2022. The secondary structure of the deleted DNA regions was predicted by RNAstructure (http://rna.urmc.rochester.edu/RNAstructureWeb/) accessed on 25 August 2022. The MetaDome web tool was used to predict the pathogenicity of identified variants, which measures the regional tolerance for genetic variation based on a missense over synonymous variant count ratio using data from the gnomAD database (https://stuart.radboudumc.nl/metadome) accessed on 18 July 2022. Protein stability was assessed by I-Mutant2.0 (https://folding.biofold.org/i-mutant/i-mutant2.0.html) and Site-Directed Mutator (SDM) (http://marid.bioc.cam.ac.uk/sdm2/prediction) accessed on 31 August 2022 using the wild-type CK2beta crystal structure (PDB: 1JWH). PSIPRED 4 (http://bioinf.cs.ucl.ac.uk/psipred) accessed on 22 August 2022 was applied to predict the secondary structure of the wild-type CK2beta and elongated mutant CK2beta (p. Met132LeufsTer110).

### 2.3. Plasmid Constructs

Human wild-type *CSNK2B* (NM_001320) and mutant (*CSNK2B*_Leu39Arg and *CSNK2B*_Met132LeufsTer110) cDNAs, as well as the *CSNK2A1* (NM_001895), were cloned into a pCMV6-Entry vector (Origene, Rockville, MD, USA). The *CSNK2B* ORFs were cloned in frame with a C-terminal Myc tag, and the *CSNK2A1* was cloned in frame with a C-terminal HA tag.

### 2.4. RNA Extraction from Blood

Total RNA was extracted from the patient’s blood samples collected in the PAXgene^®^ Blood RNA Tube via the PAXgene^®^ Blood RNA Kit (Qiagen, Hilden, Germany) and quantified with NanoDrop (ThermoFisher Scientific, MA, USA).

### 2.5. Quantitative Real-Time PCR (qRT-PCR)

Total RNA (500 ng) was reverse transcribed to cDNA using the Transcriptor First Strand cDNA Synthesis Kit (Roche, Indianapolis, IN, USA). Quantitative Real Time PCR (qRT-PCR) for *CSNK2B* mRNA was performed using standard PCR conditions in a 7900HT Fast Real Time PCR (Life Technologies, CA, USA) with the Power SYBR Green PCR Master Mix (Life Technologies, CA, USA). Gene-specific primers were designed using PRIMEREXPRESS software (Applied Biosystems, Foster city, CA). The thermal cycling conditions were composed of an initial step at 50 °C for 2 min, followed by a denaturation cycle at 95 °C for 10 min, and then 40 cycles at 95 °C for 15 s, 59 °C for 1 min and 72 °C for 30 s. The experiments were performed in biological triplicate. Expression levels were normalized to β-Actin, and all data were analyzed using the 2–ΔΔCt Livak Method [26]. The primers used in the experiments are listed in Appendix A.

### 2.6. Cell Cultures and Transient Transfection

HEK293 and HeLa cells were cultured in Dulbecco’s modified Eagles medium (Euroclone, MI, Italy) supplemented with 10% fetal bovine serum (FBS) (Euroclone, MI, Italy), penicillin 100 IU/mL and streptomycin 100 IU/mL (Euroclone, MI, Italy); the cells were maintained at 37 °C in 5% CO_2_. Cells were transfected using Fugene reagent (Promega, WI, USA) following the manufacturer’s instructions. Transfection efficiency was determined by co-transfection of a GFP-expressing plasmid.

### 2.7. Cloning PCR Product

The *CSNK2B* RT-PCR product was cloned in the vector pJET1.2 using the CloneJET PCR Cloning Kit (ThermoFisher Scientific, Waltham, MA, USA) and then transformed into competent Top10 E. Coli cells (Invitrogen, Waltham, MA, USA). Plasmids were subsequently extracted and purified with Qiagen Plasmid Mini Kit, quantified with NanoDrop (ThermoFisher Scientific, Waltham, MA, USA) and then Sanger sequenced.

### 2.8. Western Blot Analysis

The plasmids carrying the cDNA of wild-type and mutant *CSNK2B* Myc-tagged were transfected in HEK293 cells. Forty-eight hours after the transfection, cells were lysed in IPLS buffer (50 mM Tris-HCL pH7.5, 120 mM NaCl, 0.5 mM EDTA and 0.5% Nonidet P-40 (NP-40)) supplemented with protease inhibitors (Roche, Indianapolis, IN, USA). After sonication and pre-clearing, protein lysate concentration was determined by Bradford Assay (Biorad, CA, USA). Membranes were incubated with primary antibodies for 1 h at room temperature and washed in Tris-buffered saline, 0.1% Tween 20. Secondary antibodies were diluted in blocking buffer and incubated with the membranes for 1 h at room temperature. CK2beta wild-type and mutant proteins were detected using monoclonal anti-Myc antibody (9E10, 1:1000, Santa Cruz) and anti-Hsp90 monoclonal antibody (OT14C10, 1: 1000, Origene). Immunolabeled proteins were detected with the ECL Star Detection Kit (Euroclone, MI, Italy), and the images were acquired using a ChemiDoc MP imaging system (BioRad, CA, USA). The quantification of the investigated bands was achieved using ImageJ software version 1.53 t [27].

### 2.9. Fractionation Assay

After transfection, cells were washed with PBS, trypsinized and pelleted at 1100 rpm at 4 °C. The cells’ pellet was dissolved in PBS containing 0.1% Nonidet P-40 (NP-40). The cell lysate was centrifuged at 15,000 rpm for 10 s at 4 °C to separate the cytoplasmic and nuclear fractions; the supernatant was collected and boiled with SDS sample buffer (cytosolic fraction). Subsequently, the pellet was washed with 1 mL of PBS containing 0.1% NP-40 that was centrifuged, and the supernatant was discarded. The pellet was resuspended and boiled with SDS sample buffer (nuclear fraction).

### 2.10. Analysis of Protein Stability

A total of 5 × 10^5^ cells were transfected with 3.3 ug of plasmids using the Fugene transfection reagent (Promega, WI, USA). Forty-eight hours after transfection, the medium was then supplemented with 10 mM of either proteasome inhibitor MG132 (Sigma-Aldrich, MI, USA) or NH_4_CL (Sigma-Aldrich, MI, USA) for 4 h, before cell lysis in RIPA Buffer (50 mM Tris-HCl, 150 mM NaCl, 5 mM EDTA, 5 mM EGTA, 0.5% NaDoc, 1% Triton X-100 and 0.1% SDS) supplemented with protease inhibitors (Roche, Indianapolis, IN, USA). Equal amounts of proteins were resolved on 4–12% SDS-PAGE mini gels and transferred to nitrocellulose membranes (GE Healthcare, IL, USA). Western blotting was performed, as previously described, using monoclonal anti-Myc (9E10, 1:1000, Santa Cruz) and Anti-Hsp90 monoclonal antibody (OT14C10, 1:1000, Origene) to detect the wild-type and mutant CK2beta proteins and Hsp90, respectively.

### 2.11. Immunocytochemical Staining

HeLa cells were transfected using Fugene transfection reagent (Promega, WI, USA). Forty-eight hours after transfection, cells were fixed with 4% paraformaldehyde for 10 min and permeabilized with 0.5% Triton X-100 in phosphate buffered saline (PBS) for 10 min, followed by 1 h blocking in 2% BSA in PBS. Cells were then stained overnight at 4 °C with the following primary antibody diluted in blocking solution: polyclonal anti-Myc (TA150083, 1:1000, Origene). Cells were then washed with PBS and incubated for 1 h with the secondary antibody goat anti-rabbit conjugated to Alexa Fluor 594 (1:2000, Thermo-Fisher Scientific, MA, USA). The subsequent washings were performed in PBS 0.2% Tween 20. The specimens were visualized using a Zeiss fluorescence microscope.

### 2.12. Co-Immunoprecipitation

The plasmid, carrying the cDNA of wild-type or mutant *CSNK2B* Myc-tagged, was co-transfected with a plasmid expressing the CK2A1 protein HA-tagged. Forty-eight hours after transfection, the cells were lysed with the NP-40 buffer containing a protease inhibitor cocktail (Roche, Indianapolis, IN, USA), and 500 µg of total cell lysate were incubated with 15 µL of direct-conjugate immune-affinity magnetic anti-HA beads (Origene, Rockville, MD, USA) overnight at 4 °C with gentle rotation. After three washes with lysis buffer, the immunoprecipitates were resolved on 4–12% SDS-PAGE mini gels and transferred to nitrocellulose membranes (Biorad, Hercules, CA, USA). Membranes were incubated with the primary monoclonal antibodies: anti-HA (CB051, 1:1000, Origene), anti-Myc antibody (1:1000; Santa Cruz) and Hsp90 (OT14C10, 1:1000, Origene) and subsequently with the secondary anti-mouse antibody (9E10, 1:2000; Santa Cruz). Immunolabeled proteins were detected with the ECL Star Detection Kit (Euroclone, Milan, Italy), and the images were acquired using the ChemiDoc MP imaging system (BioRad, Hercules, CA, USA).

### 2.13. Immunoprecipitation and Kinase Assay

We conducted the ADP-Glo assay (Promega, WI, USA) to investigate the impact of variants on the kinase activity of the CK2 holoenzyme. The assay was set up with the casein kinase 2-purified protein (Merck, Germany), with both alpha and beta subunits, and casein kinase 2 substrate (1 ug/uL): peptide (RRRDDDSDDD). Wild-type or mutated *CSNK2B* plasmids were co-transfected with the construct expressing CK2alpha1 HA-tagged. Forty-eight hours after transfection, 500 ug of protein extracts were incubated with 30 μL of anti-Myc agarose beads (Origene, Rockville, MD, USA) overnight at 4 °C with gentle rotation. Each pre- and post-immunoprecipitation wash was performed in a centrifuge at 1000 rpm at 4 °C for 1 min with 500 μL of NP-40 complete buffer supplemented with protease and phosphatase inhibitors (Roche, Indianapolis, IN, USA). For correct assay quantification, the wild-type and mutated immunoprecipitated proteins were divided into three aliquots, one used for Western blot protein quantification, the other two for the kinase assay, in the presence and absence of substrate. According to the manufacturer’s instructions, kinase activity was measured using the ADP-GLO assay system (Promega, Madison, WI, USA).

### 2.14. Actinomycin D Treatment

HEK293 cells were transfected with plasmids carrying the cDNA of wild-type or mutant *CSNK2B* Myc-tagged. Forty-eight hours after transfection, cells were treated with 5 μM Actinomycin D (Cell Signaling Technology, MA, USA). Cells were harvested at point time *t* = 0 and *t* = 4 h, and the total RNA was extracted.

## 3. Results

### 3.1. Case Presentation and Clinical Evolution

#### 3.1.1. Patient 1

Individual 1 is an 11-year-old girl, the third child of unrelated healthy parents from Morocco with an unremarkable family history, who was previously reported in [28] (Figure 1A,C).

The proband was born at 34 weeks of gestation from a pregnancy complicated by maternal hypertension. Her birth weight was 2130 gr (−2.46 SD), and her length was 45 cm (−1.99 SD). OFC and Apgar’s scores were not available. The patient presented with post-natal growth retardation and delayed developmental milestones, including head control after 6 months and independent walking after 24 months. Examination of ectodermal tissues showed a delayed eruption of teeth, which started at 1 year, and thin hair. An arteriovenous vascular abnormality was detected in the lower right limb, which was surgically removed at 4 years. Radiological investigations revealed brain and bone malformations: a Chiari malformation with syringomyelia and a butterfly vertebra. Neurosurgical treatment for brain and spine malformations was performed at 5 years. At the same age, the girl developed epilepsy, presenting absence seizures with an EEG ictal pattern of generalized 3 Hz spike-waves complexes. She started treatment with ethosuximide but discontinued it due to gastrointestinal side effects and replaced it with valproate. At 7 years, focal seizures with impaired awareness and vomiting appeared. Multifocal epileptiform discharges were characterized the interictal EEG. Therapy with valproate and lamotrigine was effective, but compliance was discontinuous, and seizures periodically reappeared. At the last examination (11 years and 1 month), the girl presented: (1) a mild intellectual disability, mostly involving speech abilities with selective mutism; (2) a growth retardation weight of 24 kg (−2.2 SD), height 128 cm (−2.1 SD) and OFC 50 cm (−2.4 SD) and (3) epilepsy. She showed several facial dysmorphisms, including a broad forehead, frontal bossing, sparse eyebrows and thin hair, overfolded helix, hypertelorism, depressed nasal bridge, pointed chin and prognathism (Figure 1C). Tapering fingers of hands and feet and bilateral hallux valgus were noted (Figure 2A,B). The orthopantomography evidenced teeth agenesis (Figure 2C).

#### 3.1.2. Patient 2

Individual 2 is a 27-month-old boy, the third child of unrelated healthy parents from Bangladesh with an unremarkable family history (Figure 1B,D). The proband was born by caesarean section at the term of a pregnancy complicated by gestational diabetes, with a birth weight of 2630 gr (−1.49 SD), a length of 47 cm (−1.18 SD) and OFC of 34 cm (−0.83 SD). Apgar scores were 9 (1 min) and 9 (5 min). On the day of the first vaccination, at 3 months of age, the first seizure occurred triggered by fever. The EEG was normal. Shortly after, the child presented afebrile seizures characterized by bilateral clonic movements, pallor and perioral cyanosis. The EEG showed a regular background organization and the presence of bilateral synchronous paroxysmal abnormalities; a pharmacological treatment with valproic acid was started. The psychomotor development and neurological evaluation were normal. A brain MRI was normal. Subsequently, there was a significant sensibility to fever, with prolonged seizures and status epilepticus. Levetiracetam was added to therapy. At 7 months, myoclonic seizures with a head drop appeared. The EEG was characterized by diffuse paroxysmal abnormalities during sleep, on a normal background organization. The treatment with valproic acid, levetiracetam and clonazepam partially controlled the seizures. Microcephaly was detected when he was 10 months old. At the last examination (2 years 3 months), he clinically revealed: (1) mild impairment of psychomotor development; (2) low anthropometric values mostly involving the OFC (weight 11 kg (−1.45 SD), height 84 cm (−1.08 SD) and OFC 45.5 cm (−2.24 SD)); and (3) epilepsy. He presented a broad forehead with frontal bossing, hypertelorism, a depressed nasal bridge and an open mouth with a cupid’s bowed upper lip (Figure 1D).

### 3.2. Molecular Findings

WES analysis in patient 1 revealed a missense variant in exon 3 of the *CSNK2B* gene (NM_001320: c.T116G; p.Leu39Arg) (Figure 3A,B), which was classified as pathogenic according to the ACMG guidelines and Varsome (Appendix A). Sanger sequencing confirmed that the variant was heterozygous in the affected child and absent in her parents (Figure 3C, left panel).

Notably, the variant was not found in the control population when referring to several public databases (dbSNP build150, gnomAD, ExAC, Exome Sequencing Project, 1000 Genomes). No other pathogenic or likely pathogenic variants that could explain the entire or partial phenotype were detected in patient 1. More recently, the variant has been reported once in the ClinVar database (RCV002249016). We evaluated the functional impact of the substitution, using several in silico tools (PaPI score, Polyphen-2, SIFT, MutationTaster, DANN score), which predict this variant to be deleterious with a CADD Phred score of 29.5 (Table 1, Appendix A). ClustalW alignment of CK2beta orthologues revealed that the Leucine 39 is evolutionarily conserved among all species examined (Figure 3D), with a phyloP100 score of 7.376 (conservation score of the 100 species). In order to determine and corroborate putatively adverse functional effects of p.Leu39Arg, the predicted 3D structure of the mutant protein was built employing SWISS-Model using human CK2beta (PBD: 1JWH, chain C) as a template. The mutant residue (Arg) introduces a charge in a buried residue, which could result in changes in the amino acid spatial orientation, thereby leading to protein-folding problems.

The leucine 39 is located within a KEN box-like motif that consists of nine residues between amino acids 32 and 40 (Figure 3B). Although no position was entirely invariant and a minimal consensus sequence was reported, some residues are very conserved at a few positions [29]. Therefore, it is reasonable to hypothesize that the size and charge difference between the wild-type and mutant residue could compromise the KEN box-like domain structure. Mutations of single amino acids can significantly change the stability of a protein structure. Therefore, to assess whether the CK2beta stability could be influenced by the introduction of the positive-charged amino acid arginine at position 39, we used I-Mutant2.0 and SDM, which provide the Gibbs Free energy, and both predicted a decrease in protein stability upon mutation (ΔΔG= −1.84 kcal/mol and ΔΔG = −2.67 kcal/mol).

WES analysis in patient 2 identified a heterozygous deletion of 11 bp located in exon 6 of *CSNK2B*, c.384_394delAGGTGAAGCCA (NM_001320) (Figure 3A,B). Sanger sequencing confirmed that the mutation arose *de novo* in the proband (Figure 3C right panel). The variant, which was classified as pathogenic following the ACMG guidelines and Varsome and predicted as damaging by the tools used during the analysis (Appendix A), was previously described as pathogenic in an unrelated patient [28] and not found in the control population when referring to several public databases (dbSNP build150, gnomAD, ExAC, Exome Sequencing Project, 1000 Genomes). An overlapping deletion c.394-404delATGGTGAAGCT was recently identified in a patient described by Ernst and colleagues [12], suggesting the existence of a mutational hotspot. These deletion events could be caused by the putative hairpin structure of the partial palindromic DNA sequence (), which are known to form unusual secondary structures, making the surrounding DNA susceptible to genomic rearrangements [30]. Both mutations altered the frame, causing a termination site loss. The use of a new stop codon located downstream was predicted to result in both cases in a longer protein (240 instead of 215 amino acids), with an incorrect amino acid sequence compared to the wild-type CK2beta (p.Met132LeufsTer110) (Figure 4A,B), missing the zinc-finger domain and the downstream N-terminal tail where the CK2alpha interacting-region was mapped. Few hypotheses might explain the functional effect of such a mutation. The mRNA transcript comprising the canonical stop-loss allele might be degraded via mRNA surveillance pathways, leading to the haploinsufficiency of CK2beta. A second possible mechanism would make the protein unable to fold correctly. In this setting, the protein could not be very stable and might miss the zinc-finger domain and the downstream N-terminal tail, thereby losing the ability both to dimerize and to coordinate CK2alpha [31]. Third, though the mRNA surveillance pathways are, in general, quite effective at removing the mutant mRNA, with a premature stop codon/loss codon contributing, thereby, to the fine-tuning of normal gene expression, any residual transcript translated into protein could be degraded by the proteasome [32].

Further, PSIPRED 4 was applied to predict the secondary structure of the CK2beta and elongated mutant, which revealed an alteration in the number of α-helices and β-strands in the mutant polypeptide sequence compared to the wild type (Figure 5A,B), which could likely have an impact on the protein’s 3D conformation.

In patient 2, WES analysis identified an additional heterozygous *de novo* splice site mutation in the *SCN1A* gene (NM_001165963.1: c.695-1G>A), encoding the α subunit of the Nav1.1 channel. Although its functional relevance is not yet known, it might contribute to the electroclinical phenotype observed in this patient.

### 3.3. Functional Studies

#### 3.3.1. *CSNK2B* Frameshift Mutation Results in a Significantly Reduced Amount of Its mRNA

To better understand the putative causative role of p.Leu39Arg and predicted p.Met132LeufsTer110 (hereafter indicated as Leu39Arg and Met132fs, respectively), a series of in vitro experiments, ranging from gene transcription to protein translation, subcellular localization and intrinsic cellular function, were carried out. Initially, we investigated *CSNK2B* mRNA expression in the patients’ and control peripheral blood. The Quantitative Real-time PCR (qRT-PCR) showed no significant difference in *CSNK2B* expression in patient 1 (Leu39Arg) and a highly significant decrease in *CSNK2B* mRNA in patient 2 (Met132fs) compared to control mRNA expression (Figure 6A,B, respectively). In line with these results, transcript-specific qRT-PCR experiments confirmed that wild-type and mutant *CSNK2B* mRNA, amplified with transcript-specific oligonucleotides, were equally expressed in patient 1 (Figure 6C), suggesting that the transcriptional level was not altered by the mutation. In patient 2, although the mutant transcript was present (Figure 6E), qRT-PCR using primers specifically amplifying either the transcript from the wild-type allele or the mutant one revealed that the aberrant mRNA was highly significantly reduced compared to the wild-type *CSNK2B* transcript (Figure 6D). This result suggests that in patient 2, the *CSNK2B* mRNA with an 11 bp deletion, altering the frame and causing a termination site loss, is very likely unstable. To confirm this, we measured the half-life of the wild-type and mutant Met132fs *CSNK2B* mRNA in transfected cells. After treatment with actinomycin D to inhibit cellular transcription, we examined the half-life of the *CSNK2B* mRNA at different time points. The results showed that the mutant mRNA had a significantly shortened half-life compared to WT *CSNK2B* mRNA (Figure 6F,G).

#### 3.3.2. Transiently Expressed CSNK2B Missense Mutant Results in a Reduced CK2beta Protein Abundance

To gain insight into the behavior of the Leu39Arg mutant protein and of Met132fs, which were eventually translated from the residual mutant transcript, the *CSNK2B* wild type and the two mutant Myc-tagged constructs were transiently transfected in HEK293 cells. Western blot analysis on total cell lysates revealed a reduction in mutant proteins compared to the wild type (Figure 7A). Analysis of the band’s intensity indicated a significant decrease of nearly 50% in the Leu39Arg protein and an almost complete absence of the mutated Met132fs protein (Figure 7B), suggesting protein instability, as predicted by protein thermodynamic stability (ΔΔG) reduction, by PSIPRED 4 and 3D conformational changes.

#### 3.3.3. *CSNK2B* Missense Mutant Results in an Increased CK2beta Proteolysis

Since we found that in *CSNK2B* mutant-overexpressing cells, the CK2beta proteins were diminished, we investigated the molecular mechanism underlying CK2beta mutant instability. For this reason, we examined whether the reduction in CK2beta proteins was the result of degradation by treating HEK293 cells overexpressing the *CSNK2B* wild type and mutants, either with the proteasome inhibitor MG132, which effectively blocked the proteolytic activity of the 26S proteasome complex, or with NH_4_Cl, a lysosome inhibitor. Only the treatment with MG132 resulted in a significant increase in the Leu39Arg protein amount, which reached the wild-type CK2beta level (Figure 8A,B), implying that the observed reduction was related to enhanced proteolysis involving the proteasome pathway. On the contrary, the same treatment did not induce an increment in the mutant Met132fs protein to the wild-type level, which was consistent with the increased instability of the mutant transcript. In addition, the small but significant increase in Met132fs upon MG132 treatment suggests that the protein, eventually translated from the residual mutant transcript, is almost completely degraded in the cells.

#### 3.3.4. CK2beta Cellular Localization Is Not Impaired by Leu39Arg Mutation

We investigated the sub-cellular localization of the mutant protein in HeLa cells to test the hypothesis that the substitution of leucine at position 39 with an arginine Arg might alter the CK2beta biological function. The epi-fluorescence microscopy revealed a predominantly cytoplasmic localization of ectopic wild-type CK2beta and a considerably weaker signal in the case of the mutant isoform (Figure 9A). After transfection, cells were harvested, and cytosolic and nuclear protein fractions were extracted. The expression of ectopic wild-type (WT) and mutant (Leu39Arg) CK2beta proteins was analyzed by Western blotting using monoclonal anti-Myc antibody (Figure 9B lower panel), resulting in a cellular distribution pattern in line with the previous results. The same membrane was hybridized with anti-tubulin and anti-Sp1 antibodies. Tubulin and Sp1 were used as controls for cell fractionation (Figure 9B, upper and middle panels).

#### 3.3.5. Interaction with the Catalytic Subunit Is Not Impaired by Leu39Arg Mutation, but the Resulting CK2 Holoenzyme Shows a Lower Kinase Activity

Since the CK2 holoenzyme is composed of regulatory and catalytic subunits, we investigated whether the Leu39Arg could affect their binding. HEK293 cells were co-transfected with *CSNK2A1* HA-tagged with either wild-type or mutant *CSNK2B*-Myc constructs. Subsequently, pull-down assays revealed that CK2alpha1 could bind the wild-type and the mutant Leu39Arg CK2beta protein. The interaction between the catalytic subunit and the Leu39Arg mutant was decreased, likely due to its reduced protein amount (Figure 10A). We could not detect the binding with the Met132fs protein, which was nearly absent. Since Leu39Arg CK2beta and CK2alpha1 interaction was still present, but the amount of the complex was impaired, we hypothesized that this could affect the kinase activity of the holoenzyme. Upon the overexpression of *CSNK2A1* with either the wild-type or mutant *CSNK2B* construct, we performed immunoprecipitation experiments (Figure 10B), followed by a chemiluminescent specific kinase assay, in the presence or absence of the specific CK2 substrate. The data showed a significant decrease in the kinase activity of the mutant CK2beta/CK2alpha1 complex (Figure 10C).

## 4. Discussion

Poirier–Bienvenu Neurodevelopmental Syndrome (POBINDS) was recently identified as a rare neurodevelopmental disorder [11]. POBINDS is associated with heterozygous de novo mutations in the *CSNK2B* gene, which encodes for CK2beta, the regulatory subunit of the serine-threonine casein kinase 2 (CK2). The holoenzyme is a heterotetrameric complex composed of two beta regulatory and two alphacatalytic subunits. Although the CK2beta does not serve as an on/off switch of the enzyme, its primary roles are to bring the substrate protein and the catalytic subunit into proximity, close enough to facilitate the phosphorylation reaction and to stabilize the catalytic subunit [29]. CK2beta is characterized by several functional domains: the KEN box-like and the destruction domain represent putative degradation motifs; the acidic groove is involved in the modulation of the catalytic activity; the zinc-finger motif mediates the CK2beta dimerization, and finally, a C-terminal region is involved in the binding of the catalytic subunit (Figure 3B). Most published *CSNK2B* variants are missense, and the remaining are splice variants, nonsense, frameshift and small indels [13]. Initial evidence suggested that the mutations, leading to a premature stop codon, could result in a transcript likely degraded by nonsense-mediated decay (NMD) [11]. The lack of data on the effect of the other different types of *CSNK2B*-associated mutations prompted us to combine predictive functional and structural analysis and in vitro experiments to integrate the current knowledge on the pathomechanism of action of the variants, c.T116G; p.Leu39Arg and c.384_394del11; p.Met132LeufsTer110, which we identified by WES in two patients with POBINDS. In silico tools predicted the pathogenic nature of both mutations (Appendix A), supported by the high intolerance both to missense (Z score: 3.13) and the loss of function variants (pLI = 0.92) of the CSNK2B gene (gnomAD v.2.1.1.).

mRNA expression analysis from the peripheral blood RNA of patient 2 carrying the frameshift mutation, c.384_394del11, revealed that the *CSNK2B* amount was significantly less compared to the control, the transcript carrying the mutation being unstable, which was also confirmed in cells transfected with the mutant construct (Met132fs). Given that the frameshift mutation results in a stop-loss and abnormally elongated transcript, it was likely that the mutant mRNA was degraded via mRNA surveillance pathways, including No-Go and Non-Stop mRNA decay, which is activated by ribosome stalling during elongation, mainly due to defects in mRNA sequence, processing or folding [33]. In addition, our data revealed that the p.Met132LeufsTer110 protein, translated from the escaped mutant transcript, was reduced to almost undetectable levels in transfected cells, being degraded via the proteasome, likely as a consequence of an uncorrected protein folding, since the secondary structure of the extended mutant polypeptide sequence is predicted to be dramatically altered by PSIPRED 4 analysis. These data support the haploinsufficiency as an underlying pathomechanism in our patient. Six out of nine *CSNK2B* frameshift mutations known so far (including the one reported in this study) resulted in the loss of the canonical stop codon and the usage of a new termination located downstream (p.Met132Leufs*110, p.Asp155Alafs*71, p.Leu167Serfs*60, p.Pro179Tyrfs*49, p.Asn181Thrfs*46, p.Lys208Glnfs38). In line with our results, the ectopic expression of p.Leu167Serfs*60 resulted in an unstable mutant protein [34]. Conversely, the Pro179Tyrfs*49 was expressed at a level comparable to the control [35], according to the crystal structure data of the C-terminal deletion mutant CK2beta^1−182^, showing an intact global fold of the protein [5]. Nevertheless, it lost its ability to bind the catalytic subunit [35], corroborating the knowledge that the CK2beta tail (aa 181-215) plays a key role in stabilizing the β-β dimer and β-α binding [1]. It is worth noting that the mutant protein p.Met132LeufsTer110 was among these mutants, the one with the longer altered C-terminal tail, which suggests a possible correlation between the length and the magnitude of the destabilizing effect. However, further structural and functional analyses are required to prove this hypothesis.

Patient 1 had a missense mutation affecting the very conserved Leucine 39 in the KEN box-like motif (aa 32–40) in the N-terminal segment of the CK2beta (p.Leu39Arg). Notably, 30% of the published patients with the CSNK2B missense mutation and either POBINDS or IDCS presented with a recurrent or unique mutation, which significantly clustered in this short motif (Figure 11A) [12,28,34,35,36,37,38,39,40,41]. The MetaDome web tool [42], which used to interpret the significance of *CSNK2B* mutations in the tolerance landscape of the gene, indicated that the five variants were located in a region of high intolerance (low missense over synonymous variant count ratio) (Figure 11B).

As expected for this kind of mutation, the *CSNK2B* expression from the peripheral blood RNA of patient 1 was not different from the control. Indeed, the structural analysis of the mutant p.Leu39Arg suggested that the change could have a strong impact on the 3D structure, thereby leading to protein-folding problems. In this regard, stability prediction analysis revealed that this variant had the strongest destabilizing effect (lower protein thermodynamic stability (ΔΔG)) among the five known mutations in the KEN box-like motif (Table 1). Our experimental data corroborated these predictions, showing that the transiently expressed *CSNK2B* missense mutant led to a highly significant reduction in mutant CK2beta compared to the wild type, which resulted from an increased p.Leu39Arg proteolysis via the proteasome. The function of the KEN box-like domain in CK2beta has not yet been investigated, but it likely plays a role in cell-cycle-dependent protein degradation, as previously demonstrated for other proteins [43,44], and controls its own protein stability, as the Lys33 is found ubiquitinated [45], and CK2beta degraded via proteasome [7]; therefore, it is possible that other mutations in this region may also affect CK2B half-life, but functional studies are needed to prove this hypothesis.

We further observed that although the p.Leu39Arg had a sub-cellular localization similar to the wild type and was able to interact with the catalytic subunit, the amount of the complex was impaired, likely due to reduced mutant protein available.

In this condition, the resulting CK2 holoenzyme showed a significantly lower kinase activity compared to the wild type, suggesting, therefore, that the amount of CK2beta in the cells might represent a key factor, important for the fine-tuned regulation of CK2 activity. In fact, since CK2 is a constitutively active kinase, its enzymatic function has to be controlled and adjusted precisely to cope with the varying needs of a cell, assuring the correct level of phosphorylation of its numerous substrates [2]. Based on our data, p.Leu39Arg appears to behave as a LoF, thereby indicating the haploinsufficiency of CK2beta as an underlying pathomechanism in our patient.

Nevertheless, Chantalat and colleagues suggested, based on their CK2beta crystal structure, that a cluster of conserved residues composed of Asp32, Phe34 and Asn35, and Arg86, exposed at the surface, directed away from the rest of the protein, might play a role in ligands’ binding, since CK2beta acts as a docking platform [5]. Therefore, the p.Leu39Arg and other mutations in this motif could influence the 3D conformation, altering CK2beta binding properties and substrate specificity, and affecting the phosphorylation or the degradation of several substrates. However, further functional studies are required to prove this hypothesis.

In this regard, two *CSNK2B* missense mutations, the p.Asp32Asn that was originally identified in POBINDS patients, and p.Asn32His, have been recently proposed as the cause of a new intellectual disability-craniodigital syndrome (IDCS) distinct from POBINDS and characterized by cranial anomalies, facial dysmorphisms and digital anomalies. Asif and colleagues demonstrated that the two variants resulted in an increased amount of CK2beta, leading to reduced CK2 holoenzyme activity in a dominant negative manner, and thereby altering the phosphorylation of several substrates [14]. Interestingly, Zhang and colleagues, in an analysis of the correlations between mutation types and phenotypes in 57 patients, showed a statistically significant difference only in the dysmorphic features of POBINDS with non-LoF compared to LoF mutations [13]. It might be possible that the mutations located in the KEN box-like may exert a more severe dysmorphic phenotype in the patients carrying them, altering the binding properties and substrate specificity of CK2beta.

This hypothesis prompted us to investigate our patient 1’s phenotype backwards and to review those of the published patients to underline some common characteristics (Table 2).

All subjects (n = 12) showed developmental delay and cognitive and language function impairment, with a variable degree of severity. Dysmorphic features were presented in 100% of subjects, characterizing a common facial gestalt, in a range of severity and specificity [14]. Notably, our patient (p.Leu39Arg) presents a number of dysmorphic features, including prognathism, pointed chin and abnormalities of the ears, which have been described in patients carrying the p. Asn32 mutations (Table 2), creating a facial gestalt very similar to that suggested by Asif [14] as a relevant phenotypic characterization of IDCS patients. Forty-four percent of the subjects presented with either microcephaly or macrocephaly.

Skeletal abnormalities, mainly of the hands and feet, were identified in 87.5% of the patients, and the same percentage displayed variable teeth hypoplasia, hypodontia or hyperdontia. In addition, six out of seven individuals (about 86%) showed cardiac or vascular abnormalities. Of note, only 60% of the patients presented epilepsy, with a variable onset and semiology and variably associated with brain MRI abnormalities.

Despite the heterogeneity of the clinical data and the small number of patients, this comparison highlights some overlapping features, such as facial characteristics, skeletal and cardiovascular abnormalities and an abnormal number of teeth, in addition to ID, epilepsy and language impairment, and this suggests the existence of a continuous spectrum of *CSNK2B*-associated phenotypes, rather than a sharp distinction between them.

## 5. Conclusions

Our data provide in silico and functional evidence indicating that the *CSNK2B* mutations, p.Leu39Arg and p.Met132LeufsTer110, identified in two POBINDS patients, behave as LoF, which has also also been predicted for *CSNK2B* variants, resulting in a premature stop codon (PTC) [11,13,34,41]. This is suggestive of the haploinsufficiency of CK2beta as a common pathogenetic mechanism in POBINDS. However, other effects, as proposed for the p.Asp32 mutants, cannot be excluded; the reduced CK2 holoenzyme activity may represent uniform pathogenesis, in line with Asif’s results [14].

Our observations intend to be a background stimulus to promote functional analysis and to investigate the contribution of altered pathways as common neurobiological determinators of shared phenotype characteristics, leading to a continuous spectrum disorder, rather than POBINDS and IDCS as two distinct clinical entities.

The systematic collection of the facial dysmorphisms, even if mild, is suggested to confirm the existence of a common gestalt.

## Figures and Tables

**Figure 1 genes-14-00250-f001:**
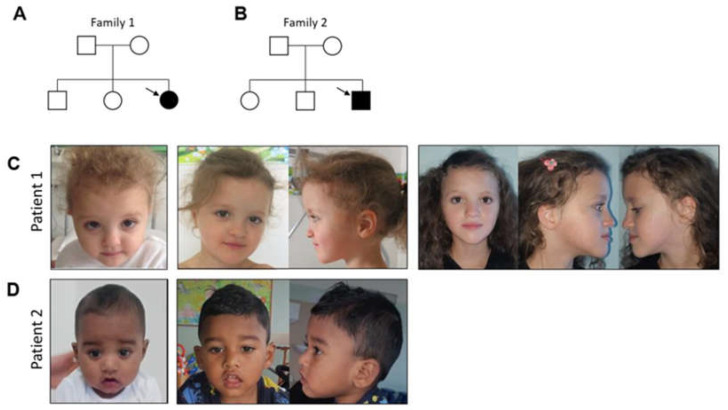
(**A**,**B**) Pedigrees of the two families included in the study. Square, male; circle, female; filled, affected; unfilled, unaffected; arrows indicate index patients. (**C**) Dysmorphic craniofacial features (front and side views) of patient 1 at 27 months (**left panel**), 5 years and 7 months (**middle panels**) and 11 years (**right panels**). (**D**) Dysmorphic craniofacial features (front and side views) presented by patient 2 at 8 months (**left panel**) and 26 months of age (**right panels**).

**Figure 2 genes-14-00250-f002:**
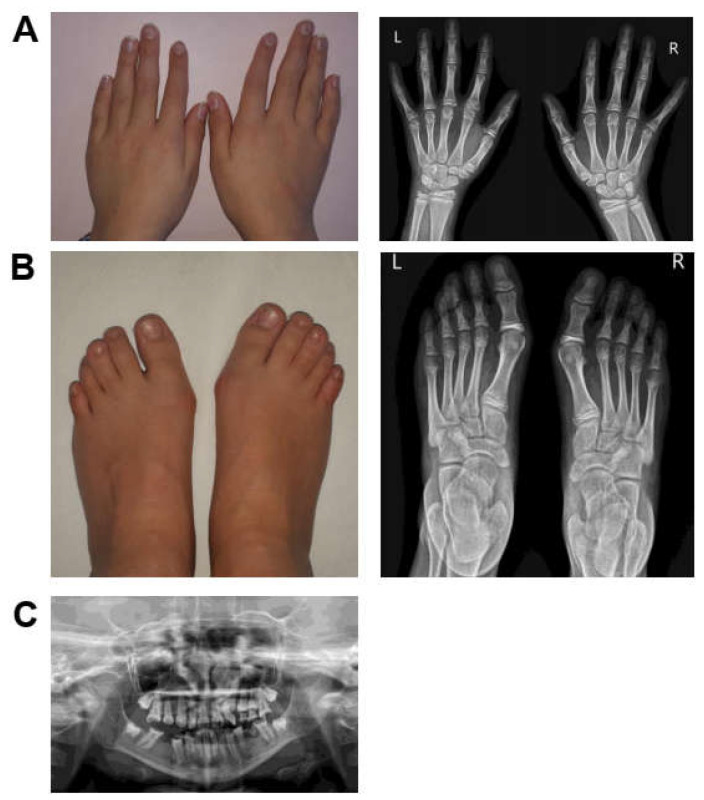
Skeletal anomalies in patient 1. (**A**) Picture and radiograph of the hands. (**B**) Picture and radiograph of the feet. (**C**) Orthopantomography of the teeth. L (left), R (right).

**Figure 3 genes-14-00250-f003:**
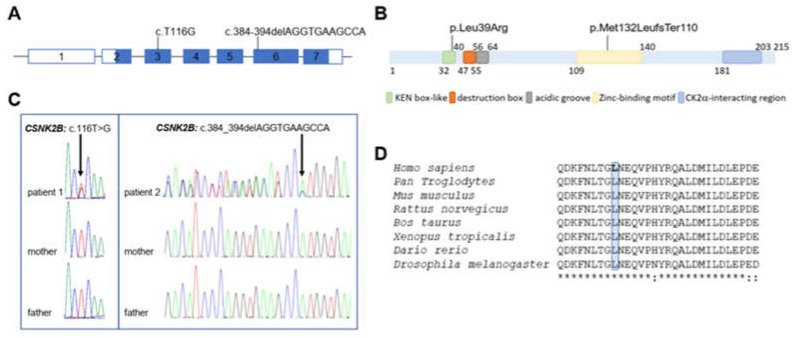
(**A**) Schematic representation of *CSNK2B* gene. The pathogenic variants identified in this study and their position are shown. Lines and boxes indicate introns and exons, respectively. Exons and introns are numbered with respect to the exon upstream of the first coding exon. The coding regions are indicated by filled boxes, whereas the non-translated sequences are depicted by open boxes. (**B**) Schematic illustration of the CK2beta protein. The position of the mutations is shown. Important regions in the CK2beta protein are indicated by filled boxes: KEN box-like (green) and destruction domain (orange) represent putative degradation motifs; the acidic groove (grey) is involved in the modulation of the catalytic activity; the zinc-finger motif (yellow) mediates the CK2beta dimerization; the region involved in the CK2alpha binding (blue). (**C**) Sanger chromatograms from patients (1–2) and their parents. Mutated nucleotides identified in patient 1 (*CSNK2B*, c.116T > G) and in patient 2 (*CSNK2B*, c.384_394delAGGTGAAGCCA) are indicated by arrows. (**D**) ClustalW multiple-sequence alignment showing conservation of Leu39 of CK2beta across species which is highlighted by the blue box. Asterisk (*) indicates conserved amino acid and colon (:) for conservative changes.

**Figure 4 genes-14-00250-f004:**
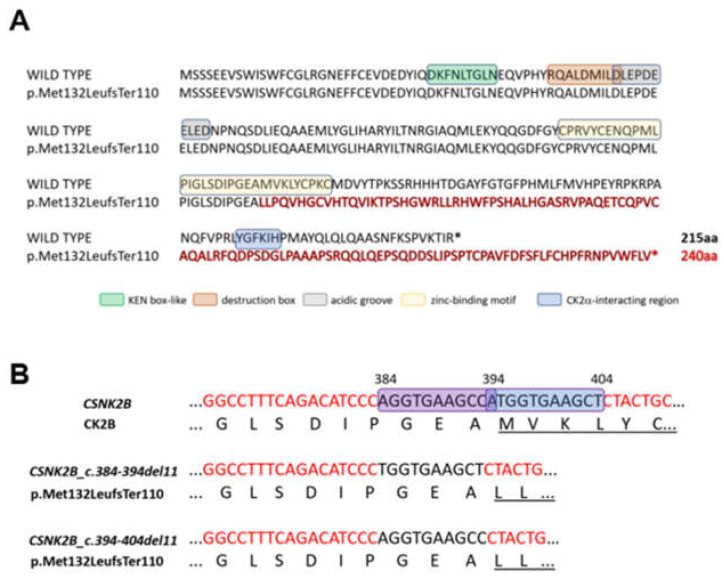
(**A**) *CSNK2B* c.384_394delAGGTGAAGCCA, identified in patient 2, is predicted to result in a longer protein (p.Met132LeufsTer110), with 240 amino acids instead of 215, and with a different C-terminal amino acid sequence (shown in red) compared to the wild-type CK2beta. * indicates the translation termination codon. (**B**) Details of the nucleotide sequences (black letters) deleted in patient 2 (current study)and in patient 19 [12] (violet and blue box, respectively). The corresponding amino acid sequences are shown and the underlined regions indicate different amino acid sequence in the mutant proteins compare to the wild type (CK2B).

**Figure 5 genes-14-00250-f005:**
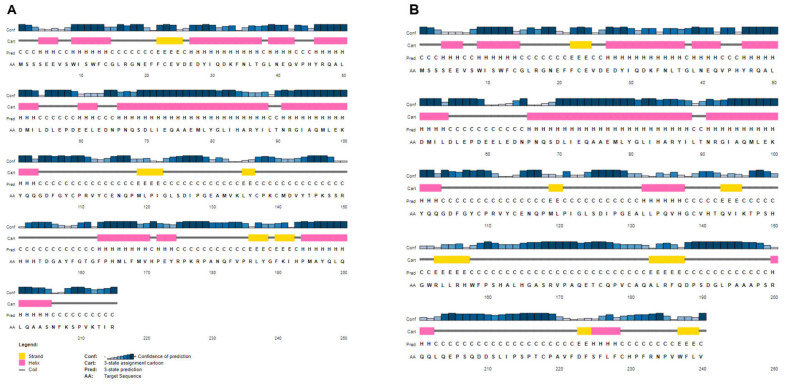
(**A**) Predicted secondary structure of the wild-type CK2beta and (**B**) of the predicted mutant elongated protein (p.Met132LeufsTer110) using PSIPRED Protein Analysis Workbench. The mutant is composed of new α-helices (pink) and strands (yellow); compare to the wild type in the predicted C-terminal tail of the mutant CK2beta protein (p.Met132LeufsTer110).

**Figure 6 genes-14-00250-f006:**
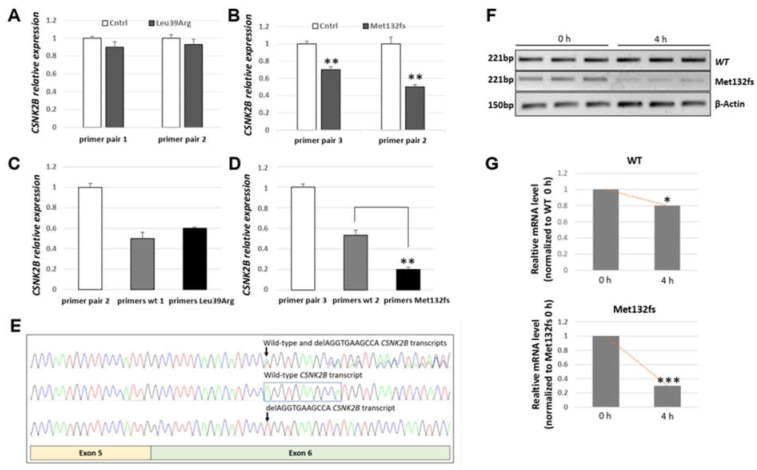
*CSNK2B* frameshift mutation reduces its mRNA stability. Quantification of *CSNK2B* transcript by qRT-PCR was performed using RNA extracted from peripheral blood of control (Cntrl) and patients (1 and 2) using primers for *CSNK2B* (pairs 1, 2 and 3) and transcript-specific oligonucleotides able to amplify either wild-type (primers wt 1, wt 2) or mutant *CSNK2B* mRNA (primers Leu39Arg and Met132fs). (**A**) Histogram showing no differences in the expression of total *CSNK2B* mRNA in patient 1 (NM_001320.7: c.116T>G, p.Leu39Arg) as compared to control. (**B**) Histogram showing a reduction of about 50% in total *CSNK2B* mRNA in patient 2 (NM_001320.7: c.384_394del11, p.Met132fs) compared to control. Each bar represents the mean ± SD of three independent experiments. ** *p* < 0.01 calculated by Student’s *t*-test (two-tailed, homoscedastic). (**C**) Allele-specific qRT-PCR in patient 1 shows a similar expression of wild-type and mutant transcripts (primers wt 1 and primers Leu39Arg, respectively). (**D**) Allele-specific qRT-PCR in patient 2 shows a significant reduction in the mutant compared to wild-type transcript (primers wt 2 and primers Met132fs, respectively). Each bar represents the mean ± SD of three independent experiments. ** *p* < 0.01 calculated by Student’s *t*-test (two-tailed, homoscedastic); Actin served as an internal control. (**E**) Sanger chromatograms of RT-PCR products using RNA extracted from patient 2. Upper panel: chromatogram showing the presence of both wild-type and mutant *CSNK2B* transcripts. Middle panel: sequence chromatogram of the wild-type and lower panel: mutant *CSNK2B* transcripts after cloning of single transcript in the pJet1.2/Blunt vector. The arrows indicate the position of the mutation. The peaks of the 11 nucleotides present in the wild-type transcript and deleted in the mutant transcript in patient 2 are highlighted with an open box. (**F**–**G**) HEK293 cells were either transfected with wild-type or mutant *CSNK2B* Myc-tagged constructs. RT-PCR assays, using primer pair 1, were performed to investigate the half-life of WT and frameshift mutant *CSNK2B* mRNAs after Actinomycin D treatment at different time points. Each bar represents the mean ± SD of three independent experiments, * *p* < 0.05; ** *p* < 0.01; *** *p* < 0.001 calculated by Student’s *t*-test (two-tailed, homoscedastic).

**Figure 7 genes-14-00250-f007:**
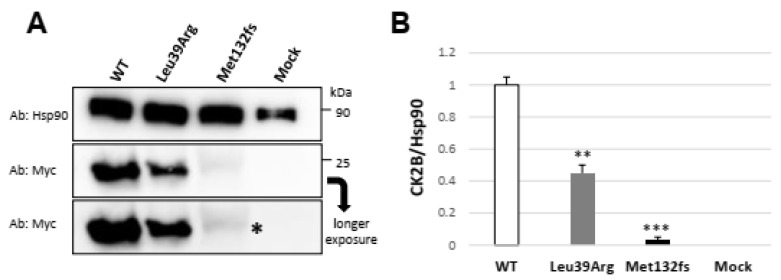
*CSNK2B* mutations impair the amount of CK2beta (CK2B). (**A**) HEK293 cells were transfected with wild-type or mutant *CSNK2B* Myc-tagged constructs. Whole-cell lysates were subjected to SDS-PAGE and immunoblotting with anti-Myc and anti-Hsp90 antibodies. A representative Western blot shows a reduced amount of mutant Leu39Arg protein compared to the wild type and an absent signal for the Met132fs mutant. After a longer exposure, a weak signal appears for the Met132fs mutant (*). (**B**) Histogram showing a significant reduction in the expression of mutant proteins compared to wild type. Each bar represents the mean ± SD of three independent experiments. ** *p* < 0.01; *** *p* < 0.001 calculated by Student’s *t*-test (two-tailed, homoscedastic). The CK2beta protein level was normalized to Hsp90 expression used as a loading control.

**Figure 8 genes-14-00250-f008:**
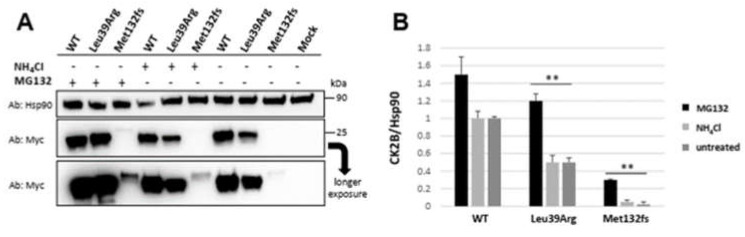
The mutant CK2beta(CK2B) protein amount is regulated by proteasome-mediated degradation. (**A**) HEK293 cells were either transfected with wild-type or mutant *CSNK2B* Myc-tagged constructs and then treated or not treated with MG132 or NH_4_Cl for 4 h. A representative Western blot shows that MG132 effectively blocks the proteolysis of mutant CK2beta proteins. (**B**) Histogram showing a significant increment in the expression of both mutant proteins upon MG132 treatment compared to untreated or treated with NH_4_Cl. ** *p* < 0.01 calculated by Student’s *t*-test (two-tailed, homoscedastic). The CK2beta protein level was normalized to Hsp90 expression used as a loading control.

**Figure 9 genes-14-00250-f009:**
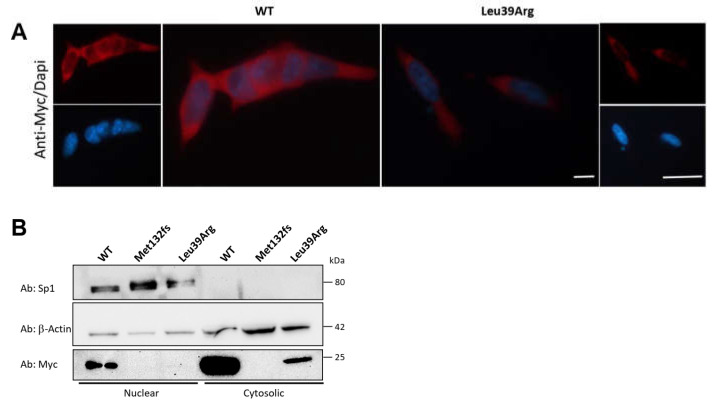
CK2beta cellular localization is not impaired by Leu39Arg mutation. (**A**) Epi-fluorescence microscopy showing the cytosolic localization of CK2beta wild type (WT) and mutant (Leu39Arg) in HeLa cells. The red signals correspond to CK2beta staining, and the blue corresponds to nuclear DAPI staining. Scale bars = 10 μM. (**B**) Nuclear and cytosolic fractions from HeLa cells transfected with wild-type CK2beta or mutants (Leu39Arg and Met132fs) were run on SDS-PAGE. The gel was blotted and probed with an anti-Myc antibody. The blot was probed with anti-β-Actin and anti-Sp1 antibodies as loading controls for cytosolic and nuclear fractions, respectively.

**Figure 10 genes-14-00250-f010:**
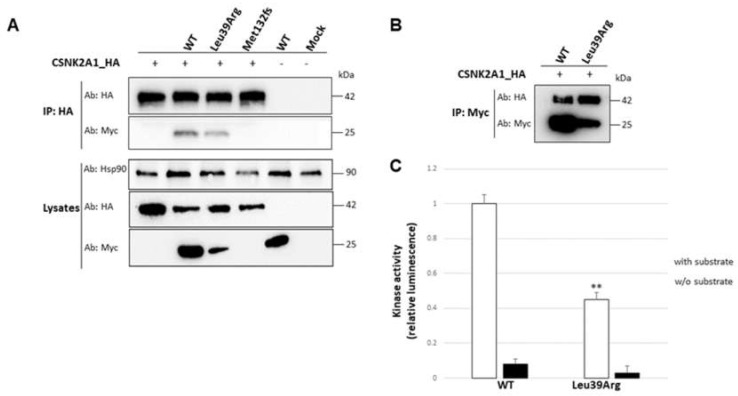
Impact of *CSNK2B* mutations on CK2 subunits’ interaction and kinase activity. (**A**) HEK293 cells were transiently co-transfected with *CSNK2A1*-HA and the Myc-tagged *CSNK2B* wild type and mutants. Protein complexes were immunoprecipitated using an anti-HA antibody. Proteins were subjected to SDS–PAGE gel and immunoblotted with an anti-Myc antibody and an anti-HA antibody. Total cell lysates from co-transfected and non-transfected cells (mock) were probed with anti-Myc, anti-HA and anti-Hsp90 as a loading control. (**B**) Measure of the kinase activity of wild type (WT) and mutant (Leu39Arg) CK2beta proteins. *CSNK2B* wild-type and mutant constructs were overexpressed with the catalytic subunit *CSNK2A1*-HA, followed by immunoprecipitation with anti-Myc antibody and luminescent ADP-Glo kinase assay with (+) and without (−) CK2 specific substrate. (**C**) Histogram showing a significant decrease in the kinase activity of the mutant complex compared to the wild type. Experiments were repeated three times with independently prepared cell extracts. ** *p* < 0.01 calculated by Student’s *t*-test (two-tailed, homoscedastic).

**Figure 11 genes-14-00250-f011:**
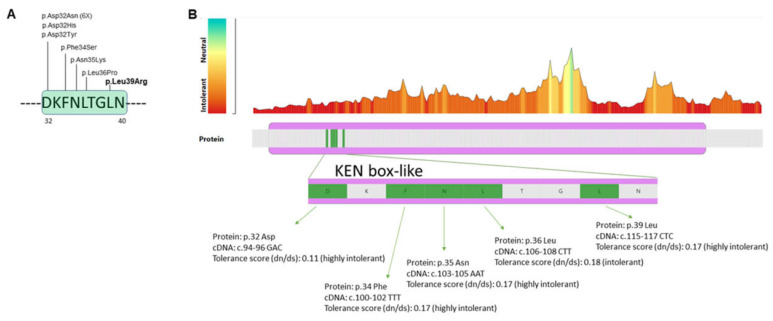
(**A**) Recurrent or unique missense mutations reported so far (HGMD database and current study) affecting the KEN-box-like domain (aa 32-40) of the CK2beta protein (NP_001311.3). The p.Leu39Arg identified in patient 1 is indicated in bold. (**B**) MetaDome web server for the CSNK2B gene. The tolerance landscape (upper panel) indicated the region that is intolerant to missense variation. Enlargement of the KEN box-like motif located between protein positions 32 and 40. The mutations identified in patients with IDCS or POBINDS are indicated in green. The analysis shows that these mutations are located in a region of high intolerance. The non-synonymous over-synonymous ratio, or dn/ds score, is used to quantify genetic tolerance [42].

**Table 1 genes-14-00250-t001:** Prediction tools were used to interpret the possible deleterious effect of CK2beta mutations affecting the KEN box-like motif. I-Mutant2.0 and SDM were used to predict protein stability upon the mutation; they provide the folding free energy and the stability difference score (ΔΔG). A ΔΔG ≥ 0.0 corresponds to mutations predicted to be stabilizing, whereas ΔΔG <0.0 corresponds to mutations predicted to be destabilizing.

CK2B Mutation (NP_001311.3)	p.Asp32Asn	p.Asp32His	p.Asp32Tyr	p.Phe34ser	p.Asn35Lys	p.Leu36Pro	p.Leu39Arg
**CADD score**	28.9	29.7	32	29.4	23.8	28.9	29.5
**I-Mutant2.0**	ΔΔG= −0.49	ΔΔG= −0.97	ΔΔG= −0.28	ΔΔG= −2.74	ΔΔG= −2.42	ΔΔG= −2.55	ΔΔG= −2.32
**SDM**	ΔΔG= −0.21	ΔΔG= −0.01	ΔΔG= −0.05	ΔΔG= −0.39	ΔΔG= −0.98	ΔΔG= −2.29	ΔΔG= −2.67

**Table 2 genes-14-00250-t002:** Main clinical features of the patients with CSNK2B missense mutations affecting the KEN box-like. +: present; - not present; NA: not available.

Reference	Patient 2 [14]	Patient [14]	Patient 7 [14] Previously Patient 25 [12]	Patient 8 [12]	Patient 14 [12]	Patient 2691 [38]	Patient 1 [14]	Patient 2 [39]	Patient 10 [12]	Patient 12 [12]	Patient 48 [35]	Patient 1 Current Study	Total
**Mutation (NP_001311.3)**	p.Asp32Asn	p.Asp32Asn	p.Asp32Asn	p.Asp32Asn	p.Asp32Asn	p.Asp32Asn	p.Asp32His	p.Asp32Tyr	p.Phe34Ser	p.Asn35Lys	p.Leu36Pro	p.Leu39Arg	
**Gender**	F	F	M	M	M	M	F	M	F	M	M	F	5F:7M
**Age (years; months)**	10y	2y	13y	12y	31y	2y (died)	19y	5y	17y	19m	11y 5m	11y	19m-19y
**Head abnormalities**	-	-	-	NA	NA	microcephaly (HP:0000252)	-	Microbrachy-cephaly	-	macrocephaly (HP:0000256)	NA	microcephaly (HP:0000252)	4/9 (44.4%)
**Craniofacial features**	asymmetrical ears (HP:0010722); deep-set eyes (HP:0000490); hypoplastic alae nasi (HP:0000430); thin upper lip (HP:0000219); prognathism (HP:0000303); pointed chin (HP:0000307)	deep-set eyes (HP:0000490); hypoplastic alae nasi (HP:0000430); prognathism (HP:0000303)	anteverted ears (HP:0001249); upslantong palpebral fissures (HP:0000582); bulbous nose (HP:0000414); smooth philtrum (HP:0000319); thin upper lip vermilion (HP:0000219)	mild dysmorphic features	mild dysmorphic features	External ear anomaly (HP:0000356); thick eyebrows (HP:0000574); short neck (HP:0000470)	small ears (HP:0008551) with prominent antitragus; deep-set eyes (HP:0000490); broad nasal bridge (HP:0000431); hypoplastic alae nasi (HP:0000430); thin upper lip (HP:0000219); prognathism (HP:0000303); pointed chin (HP:0000307)	prominent and dyplastic ears (HP:0000411; HP:0000377); low nasal bridge (HP:0005280); anteverted nares (HP:0000463); malar hypoplasia (HP:0000272); high and narrow palate (HP:0002705)	mild dysmorphic features	mild dysmorphic features	NA	overfolded helix (HP:0000396); broad forehead (HP:0000337); frontal bossing (HP:0002007); sparse eyebrows (HP:0045075), hypertelorism (HP:0000316); depressed nasal bridge (HP:0005280); prognathism (HP:0000303); pointed chin (HP:0000307)	11/11 (100%)
**Ectodermal anomalies**	thin hair (HP:0008070); small teeth (HP:0000691)	sparse hair (HP:0008070); coarse hair (HP:0002208); delayed teeth eruption (HP:0000684)	supernumerary tooth (HP:0011069)	NA	NA	-	thin hair (HP:0008070); hypodontia (HP:000668)	duplicate upper right incisor (HP:0011064); nail hypoplasia (HP:0001792)	NA	NA	NA	thin hair (HP:0008070); delayed teeth eruption (HP:0000684); teeth agenesis (HP:0009804)	6/7 (85.7%)
**Skeletal abnormalities**	tapered fingers (HP:0001182); bilateral clinodactyly of toes (HP:0001863)	brachydactyly (HP:0001156); contractures of the hand (HP:0009473)	brachydactyly (HP:0001156)	NA	NA	-	bilateral syndactily of the fingers (HP:0010492) and toes (HP:0001770); clinodactyly (HP:0001863); broad thumb (HP:0011304)	short fifth fingers (HP:0009237); tapered distal phalanges of fingers (HP:0009884); left hand postaxial polydactyly (HP:0100259)	unilateral polydactily (HP:0010442)	NA	NA	tapered fingers (HP:0001182); hallux valgus (HP:0001822); butterfly vertebrae (HP:0003316)	7/8 (87.5%)
**Short stature (HP:0004322)**	-	-	-	NA	NA	+	+	NA	NA	NA	NA	+	3/6 (50%)
**Developmental Delay (HP:0001263)**	+	+	+	+	+	+	+	+	+	+	NA	+	11/11 (100%)
**Intellectual disability (HP:0001249)**	+	NA	mild-moderate	moderate (HP:0002342)	moderate (HP:0002342)	mild (HP:0001256)	moderate (HP:0002342)	NA	mild (HP:0001256)	NA	NA	mild (HP:0001256)	8/8 (100%)
**Speech impairment (HP:0002167)**	+	+	dyspraxia (HP:0011098)	+ ( 25 words at 5)	+ (few words)	+	dysarthria (HP:0001260)	NA	delayed speech (HP:0000750)	does not babble	NA	Delayed speech and language development (HP:0000750); abnormal verbal communicative behavior (HP: 4000072)	10/10 (100%)
**Seizures (HP:0001250)**	- (EEG abnormalities)	-	+ (absences; onset: 7y)	+ (febrile sz; onset: 1.5m)	+ (absences, tonic-spasms; onset: 2y)	-	+ (tonic-clonic; onset: NA)	+ (onset: 9 m) muscular tone loss abnormal eyes movements	-	-	+ (onset: NA)	+ (absences; onset: 5y)	7/12 (58.3%)
**Abnormal brain MRI**	NA	NA	-	(2y) mild to moderate diffuse abnormal signal in the cerebral white matter (HP:0002500); low volume ventral pons	NA	-	NA	NA	periventricular gliosis	NA	NA	Chiari malformation (HP:0002308); syringomyelia (HP:0003396)	3/5 (60%)
**Cardiac/** **vascular abnormalities**	WPW (HP:0001716)	fenestrated atrial septal defect (HP:0001631)	patent ductus arteriosus (HP:0001643)	NA	NA	-	Ebstein’s anomaly (HP:0010316) atrial septal defect (HP:0001631)	NA	NA	patent foramen ovale (HP:0001655)	NA	arteriovenous vascular abnormality (HP:0100026)	6/7 (85.7%)

## Data Availability

The data presented in this study are available upon request from the corresponding author. The data are not publicly available due to privacy restrictions.

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
