# Peer review of "Haploinsufficiency as a Foreground Pathomechanism of Poirer-Bienvenu Syndrome and Novel Insights Underlying the Phenotypic Continuum of CSNK2B-Associated Disorders"

_genes, 2023, doi:10.3390/genes14020250_

Round 1

Reviewer 1 Report

The present study seems to lead to a conclusion in functional evidence in the recurrent association between mutations in CSNK2B gene and POBINDS. Here, authors identify two probable de novo mutations in two probands with POBINDS phenotype in the same gene. Both variants are categorized as LoF, so, due to the scarce knowledge in functional effect of variants in this candidate gene, they perform predictive and functional analysis. Functional analysis is an important step in the research of any rare disorder, as several associated mutations usually lack of enough replication on other similar patients due to how rare the phenotype may be. This is a strong deal in this study.

However, I have some questions regarding the article, that may be adressed easily:

1) Lack of enough info for replicability in the next parts: DNA extraction, bioinformatic tools used for aligmnent, calling and priorization, instead to enforce the reader to look for a cited reference. In the text is mentioned Supplementary Data, but it is better to add tools, versions and the different parameters used for the tools in order to provide a better understanding for possible replication.

2) Some images lack of enough quality (perhaps due to the compression of the pdf?). Fig 12 for example is quite pixelated and it is difficult to read variant annotations. Also Fig 6 bar legends are a bit blurred and the text size is a bit small.

3) Fig 4 is merely informative, as I understand. I suggest to add the predictive effects on the protein to the figure, not only the change of aminoacid. This should be mandatory as the folding modification noted in the text is barely visible in the representation and it is a complete assumption.

Author Response

Point 1: Lack of enough info for replicability in the next parts: DNA extraction, bioinformatic tools used for aligmnent, calling and priorization, instead to enforce the reader to look for a cited reference. In the text is mentioned Supplementary Data, but it is better to add tools, versions and the different parameters used for the tools in order to provide a better understanding for possible replication.

Response 1: We thank the Reviewer for the comments. We have added the requested information in the text in order to provide a better understanding for possible replication.

Point 2: Some images lack of enough quality (perhaps due to the compression of the pdf?). Fig 12 for example is quite pixelated and it is difficult to read variant annotations. Also Fig 6 bar legends are a bit blurred and the text size is a bit small.

Response 2: We have modified the figures to higher resolution images in order to improve readability.

Point 3: Fig 4 is merely informative, as I understand. I suggest to add the predictive effects on the protein to the figure, not only the change of aminoacid. This should be mandatory as the folding modification noted in the text is barely visible in the representation and it is a complete assumption.

Response 3: We thank for the comments. We agree with the Reviewer that the folding modification in the 3D structural model, built employing the Swiss-model, a fully automated protein structure homology-modelling server, might result barely visible in this representation. Since the Figure 4 is merely descriptive, as correctly pointed, we removed it, maintaining the assumption in the text to provide more clarity for the reader.

Reviewer 2 Report

Di Stazio and colleagues identified pathogenic variants of CSNK2B in two unrelated individuals manifesting POBINDS. The authors performed several assays and elegantly revealed the functional consequences of the identified variants and suggested haploinsufficiency as an underlying pathomechanism in studied patients.  

This manuscript is well written and the results are discussed appropriately. Though the results described in this manuscript are potentially interesting, this reviewer thinks to compare the facial gestalt of the CK2B;p.Leu39Arg patient with those carrying CK2B;p.Asp32. It will help improve the concluding remarks where the author proposed that patients carrying variants in the KEN box-like motif might suggest the existence of a continuous spectrum of CSNK2B-associated phenotypes.  

Minor Suggestions

Figure 8A. It would be more informative if the Hsp90 blot moved to the bottom of the figure, beneath the Myc blots.

Figure 8B. The Y-axis of the bar graph shows ‘Hsp90/CK2β’ which is confusing for the readers. Please explain it in the legend.

Figure 10: IF of Met132fs is not shown.

Please add “beta” at the end of the protein encoded by CSNK2B throughout the manuscript  

Line 665: ‘which activated by ribosome stalling during…..’ Correction: which ‘is’ activated by the ribosome stalling during……

Page 16, line 676, please add p. before Asn181Thrfs*46, Lys208Glnfs38

Line 717: …. among those five known mutations in the KEN box-like. Please add ‘motif’ at the end of this sentence.

Author Response

Point 1: This manuscript is well written and the results are discussed appropriately. Though the results described in this manuscript are potentially interesting, this reviewer thinks to compare the facial gestalt of the CK2B;p.Leu39Arg patient with those carrying CK2B;p.Asp32. It will help improve the concluding remarks where the author proposed that patients carrying variants in the KEN box-like motif might suggest the existence of a continuous spectrum of CSNK2B-associated phenotypes.  

Response 1: We are grateful to the Reviewer for the comment and we have implemented the suggestion in the manuscript.  

We agree with the Reviewer that the comparison of the facial features of the patient with p.Leu39Arg with those carrying mutations affecting the asparagine 32 highlights a common facial gestalt and enforces the suggested existence of a continuous spectrum of CSNK2B-associated phenotypes. 

Point 2: Figure 8A. It would be more informative if the Hsp90 blot moved to the bottom of the figure, beneath the Myc blots.

Response 2: We thank the Reviewer for the comment. As usually done and for consistency, all blots are shown according to molecular weight (kDa) of the investigated proteins, from higher to lower one.

Point 3: Figure 8B. The Y-axis of the bar graph shows ‘Hsp90/CK2β’ which is confusing for the readers. Please explain it in the legend.

Response 3: We thank the Reviewer for the comment. We have added the requested information in the legend.

Point 4: Figure 10: IF of Met132fs is not shown.

Response 4: IF of Met132fs is not shown since the signal is not detectable due to the extremely low Met132fs protein amount.

Point 5:

Please add “beta” at the end of the protein encoded by CSNK2B throughout the manuscript  

Line 665: ‘which activated by ribosome stalling during…..’ Correction: which ‘is’ activated by the ribosome stalling during……

Page 16, line 676, please add p. before Asn181Thrfs*46, Lys208Glnfs38

Line 717: …. among those five known mutations in the KEN box-like. Please add ‘motif’ at the end of this sentence.

Response 5: We have done the requested changes.

Reviewer 3 Report

 Excellent research work, very well described but somewhat long.

Although facial dysmorphia are mild and do not constitute an important factor in the diagnosis in POBINDS, they should be better described.

Figure 4 is superimposed on the text.

In summary, excellent work.

Author Response

Point 1: Although facial dysmorphia are mild and do not constitute an important factor in the diagnosis in POBINDS, they should be better described.

Response 1: We are grateful to the Reviewer for the comments on this relevant issue. We have implemented the Reviewer’s suggestion in the manuscript.  

We agree with the Reviewer that currently in the diagnosis of POBIND, facial features do not constitute an important factor since not always sufficient accuracy is paid to their description. Nevertheless, a continuous spectrum of CSNK2B-associated phenotypes might exist, therefore a systematic collection of facial dysmorphisms, even if mild, might be relevant to define a common facial gestalt.

Point 2: Figure 4 is superimposed on the text.

Response 2: It has been changed.